evolution, genetics, ecology

climate change, genetic divergence, niche model, pied flycatchers, *ficedula*, last glacial maximum

**Author for correspondence:**
Vera M. Warmuth
e-mail: warmuth@bio.lmu.de

# Major population splits coincide with episodes of rapid climate change in a forest-dependent bird

Vera M. Warmuth[1,2], Malcolm D. Burgess[3,4], Toni Laaksonen[5], Andrea Manica[6], Marko Mägi[7], Andreas Nord[8], Craig R. Primmer[9,10], Glenn-Peter Sætre[11], Wolfgang Winkel[12] and Hans Ellegren[2]

[1]Department of Evolutionary Biology, Biozentrum Martinsried, Ludwig-Maximilians Universität München, Planegg-Martinsried, Germany
[2]Department of Evolutionary Biology, Evolutionary Biology Centre (EBC), Uppsala University, Uppsala, Sweden
[3]Centre for Animal Behaviour, University of Exeter, Exeter, UK
[4]RSPB Centre for Conservation Science, Sandy, UK
[5]Department of Biology, University of Turku, Turku, Finland
[6]Department of Zoology, University of Cambridge, Cambridge, UK
[7]Institute of Ecology and Earth Sciences, University of Tartu, Tartu, Estonia
[8]Department of Biology, Section for Evolutionary Ecology, Lund University, Lund, Sweden
[9]Organismal and Evolutionary Biology Research Program, and [10]Institute of Biotechnology, Helsinki Institute of Life Sciences (HiLIFE), University of Helsinki, Finland
[11]Centre for Ecological and Evolutionary Synthesis, University of Oslo, Oslo, Norway
[12]Institute of Avian Research, 'Vogelwarte Helgoland', Wilhelmshaven, Germany

VMW, 0000-0002-0305-5125; AN, 0000-0001-6170-689X; CRP, 0000-0002-3687-8435; G-PS, 0000-0002-6236-4905; HE, 0000-0002-5035-1736

Climate change influences population demography by altering patterns of gene flow and reproductive isolation. Direct mutation rates offer the possibility for accurate dating on the within-species level but are currently only available for a handful of vertebrate species. Here, we use the first directly estimated mutation rate in birds to study the evolutionary history of pied flycatchers (*Ficedula hypoleuca*). Using a combination of demographic inference and species distribution modelling, we show that all major population splits in this forest-dependent system occurred during periods of increased climate instability and rapid global temperature change. We show that the divergent Spanish subspecies originated during the Eemian–Weichselian transition 115–104 thousand years ago (kya), and not during the last glacial maximum (26.5–19 kya), as previously suggested. The magnitude and rates of climate change during the glacial–interglacial transitions that preceded population splits in pied flycatchers were similar to, or exceeded, those predicted to occur in the course of the current, human-induced climate crisis. As such, our results provide a timely reminder of the strong impact that episodes of climate instability and rapid temperature changes can have on species' evolutionary trajectories, with important implications for the natural world in the Anthropocene.

## 1. Introduction

Climate change can promote lineage divergence by introducing vicariant barriers that result in altered patterns of gene flow, hybridization and selection across the landscape [1]. The Quaternary period (2.6 million years—present) is known for its dramatic climatic fluctuations. Composed of the Pleistocene and the current Holocene Epochs, the Quaternary has been marked by more than 50 climate oscillations of various magnitudes and durations as well as 8–10 major glaciations during the past 800 thousand years alone [2]. It is now well established that Quaternary climate change has profoundly impacted on

today's biota, causing widespread extinctions in some taxa and promoting lineage diversification in others, thereby shaping global patterns of biodiversity [3–5].

Ice core data, together with other sources of information on past climates and environments have led to a detailed characterization of Quaternary climate cycles, each of which is now known to have had its own idiosyncrasies in terms of the timing and magnitude of changes [6–8]. Among the most dramatic Quaternary periods in terms of their perceived impact on Northern Hemisphere biota is the last glacial maximum (LGM, 26.5–19.0 kya, [9]), the most recent interval in Earth history when global ice sheets reached their maximum volume. Indeed, a handful of high latitude and alpine taxa diverged during the LGM, most likely as a direct consequence of advancing glaciers [10–12]. However, in most cases, large confidence intervals around divergence time estimates from genetic data make it impossible to link species divergence to such narrow intervals, let alone individual climate cycles. Ultimately, these uncertainties prevent us from assessing the impact of different climate change metrics (velocity, duration, geographic scale, magnitude) on past, and potentially future, biota.

Accurate estimates of the rate at which mutations accumulate are key to the accurate inference of common ancestry dates within and among species [13]. Whole-genome sequence data from multiple parent–offspring pedigrees offer a direct means of obtaining mutation rate estimates (pedigree-based approach). Direct mutation rates have the potential to yield lower error divergence time estimates at or below the species level than mutation rates calculated indirectly [14,15]; however, challenges associated with sequencing family pedigrees means that direct mutation rates currently exist for only a handful of vertebrate taxa (primates: [14,16,17]; wolves [18]).

Here, we make use of the first direct estimate of the germ-line mutation rate for a passerine bird—the collared flycatcher (*Ficedula albicollis*) [19]—to infer the timing of intra-specific diversification events in its sister taxon, the pied flycatcher (*F. hypoleuca*). The oldest within-species diversification events in temperate avian taxa have been dated to around 1 million years ago [20,21]. For relatively recent timescales such as these, direct mutation rates promise to yield more accurate molecular dates than phylogenetic mutation rates, especially in birds, where fossil preservation is poor, and calibrations from distantly related taxa compromise estimates of divergence times from phylogenetic trees [22]. In conjunction with species distribution modelling, we further test the widely held assumption that pied flycatchers survived the LGM in the Iberian Peninsula and expanded across Europe at the end of the last glacial period [23,24].

Species distribution models (SDMs) have emerged as important tools in evolutionary studies, as they can be used to derive spatially explicit predictions of environmental suitability for species under past climatic conditions. This is typically achieved by statistically relating (bio-) climatic variables to species occurrence or abundance data [25,26]. SDMs have been used to study bird distributions in a variety of contexts, including late Pleistocene niche reconstruction [27–29].

## 2. Material and methods

### (a) Samples

We follow molecular-based taxonomies of the genus *Ficedula*, whereby *F. h. speculigera*, formerly regarded as a subspecies of pied flycatchers, is a full species, *F. speculigera* (Atlas flycatcher,

e.g. [20,30]), and whereby the only population genetically distinct enough from the nominal *F. h. hypoleuca* to warrant sub-species status is the Spanish *F. h. iberiae* [31]. See the electronic supplementary material for a more detailed discussion of the *F. hypoleuca* nomenclature. Pied flycatcher samples from nine localities distributed throughout western Europe were included in this study (electronic supplementary material, figure S1 and table S1). Samples from Latvia, Estonia, southern Sweden (Lund), Norway, the UK and Germany represent a subset of those included in [32] and were extracted as described in that study. Samples from the UK were provided as whole blood samples preserved in ethanol; these were extracted following standard phenol–chloroform extraction. Samples from Spain, the Czech Republic and eastern Sweden (Uppsala) represent a subset of those included in [33] and were available as DNA extracts. For details of the extraction protocol used see [33]. Unfortunately, we do not have samples from the eastern Eurasian part of the pied flycatcher breeding range. While this might mean that major population splits in eastern Eurasia might go undetected, it will not affect the conclusions we can draw for western European pied flycatchers.

### (b) RADSeq data

Double-digest restriction-associated DNA sequencing (ddRAD-Seq) libraries were generated using a modified version of the protocol by [34]. In brief, we digested DNA with two different restriction enzymes, *MseI* and *SbfI*, and ligated adaptors containing unique 6-base nucleotide barcodes to the digested genomic fragments such that each individual received a unique barcode. We then amplified the barcoded restriction-ligation products using PCR and standard Illumina primers. We ran four replicate PCRs for each individual and then pooled the four PCR replicates to yield a single PCR pool per individual. Using gel electrophoresis, we size-separated DNA fragments and excised 300–500 base pair fragments from the gel. We used a QIAquick Gel Extraction kit (Qiagen Inc.) to purify gel punches and pooled individual libraries for sequencing. Libraries were sequenced on an Illumina HiSeq2500 instrument using paired-end sequencing (150 bp).

Small fragment sizes can be carried through the size selection step of ddRADSeq protocols, causing the sequencer to read through the restriction enzyme cut site on the 3′ end into the P2 adapters [35]. Thus, we removed Illumina's P1 and P2 adaptors from both 5′ and 3′ ends of the fragments using Cutadapt 1.9 [36]. Overlapping reads were then merged using FLASH [37] under default settings and retaining only reads with a minimum final length of 50 bp. Both merged reads and unmerged read pairs were then demultiplexed using the *process_radtags* module of Stacks [38] and mapped to the *F. albicollis* reference genome using the mem algorithm implemented in Burrows–Wheeler Alignment Tool (BWA, [39]).

### (c) Population genetic analyses

We have previously shown that inferring allele frequencies directly from RADSeq data produces less biased estimates than allele frequencies inferred from called genotypes [40]. We therefore estimated allele frequencies without first calling genotypes using the maximum-likelihood method described in [41] and implemented in ANGSD v. 0.913 [42]. The genotype likelihoods required by this method were calculated using the GATK model (GL = 2), and both major and minor alleles were inferred from genotype likelihoods (doMajorMinor = 2). We discarded reads with multiple hits (minMapQ 1) and adjusted mapping quality for excessive mismatches (C = 50). To minimize the impact of selection on demographic inference, we removed reads that overlapped with known coding regions, conserved elements [43] and regions of high CpG content [44,45] using the intersect function of BEDTools v. 2.26.0 with a minimum overlap of 1 bp. To identify a list of high-confidence variable sites, we computed standard per-BASE Alignment Quality (BAQ = 1) according to [46] and set

a minimum base quality threshold of 20 (minQ = 20). We further required variable sites to be present in a minimum of 60 out of the 90 individuals, have a *p*-value below $10^{-6}$, and a minor allele frequency (MAF) above 0.05. The resulting set of variable sites was used for all downstream analyses.

We used principal component analysis (PCA, a non-parametric approach) to identify the major axes of genetic variation in our dataset and admixture analysis (a model-based approach) to allocate individuals into discrete populations based on their admixture proportions. PCA was conducted using PCAngsd version 0.986 [47] and we used NgsAdmix [48] for admixture analysis. All plots were generated using R v. 4.0.3.

## (d) Demographic inference

We used DADI v. 1.7.0 [49] to estimate divergence times between pairs of populations representing the three major pied flycatcher sub-populations in our dataset: Spanish pied flycatchers (*F. h. iberiae*), a differentiated UK lineage and a weakly structured population including birds sampled across western Europe and Scandinavia. Split times involving the latter group were estimated using three different subgroups to test for effects of weak sub-structuring on time estimates: Germany (GEL), Czech Republic (CZR) and Latvia (LAT). The choice of these subgroups was based on evidence from admixture analysis showing slight clustering of birds sampled at each of these locations (electronic supplementary material, figure S2).

For a given demographic model, DADI extracts the expected site frequency spectrum (SFS) and calculates the composite likelihood between the expected and observed SFS. The demographic model that was previously shown to best fit pied flycatcher genetic data is one of a population size change in the ancestral population, followed by a split with migration [40]. To estimate split times between the three major genetic clusters, we generated two-dimensional SFS between pairs of populations representing these clusters in ANGSD and ran DADI using the two-population model described above and in [40]. We additionally ran a three-population model with a three-dimensional SFS generated from the Spanish, UK and Czech populations. The three-population model was analogous to the two-population model, but included two successive splits with migration, rather than one split, as in the two-population scenario.

In DADI, the (population) mutation parameter theta is given in units of $4 * N_{ref} * \mu$, where $N_{ref}$ denotes a population at equilibrium and of non-zero size and $\mu$ is the per-base mutation rate of the study taxon. $\mu$ is calculated as the mutation rate times the length $L$ of the sequence from which SNPs are derived. $L$ is estimated from the data as

$$L = L_{unfiltered} \times \frac{\text{segregating sites in the filtered dataset}}{\text{segregating sites in the unfiltered dataset}}. \quad (2.1)$$

Times are given in units of $T = 2 * N_{ref}$ generations, and migration rates are given in units of $M_{ij} = M_{ji} = 2 * N_{ref} * m$. Using the above relationships, DADI parameters can then be converted to real values. We used the recently published direct estimate of the mutation rate for the closely related collard flycatcher of $2.3 \times 10^{-9}$ mutations per site per year [19] and a generation time of two years [50]. Mutation rates can differ even between closely related taxa; however, differences between the closely related collared and pied flycatchers are unlikely to exceed variation found across the hominins, whose evolutionary history is studied using direct mutation rates obtained from modern humans [51].

## (e) Linking climate and population history

To assess the role of past climate fluctuations in driving population divergence, we linked the divergence times of the major pied flycatcher lineages to global climate system change using Antarctic temperature estimated from the high-resolution deuterium profile of the EPICA (European Project for Ice Coring in Antarctica) Dome C ice core as a proxy. Antarctic temperature is highly correlated with average global temperature [52]. Rates of change (ROC) were calculated over 1000 year-intervals as

$$\text{ROC} = \frac{T - \text{lag}(T)}{\text{lag}(T)} \times 100, \quad (2.2)$$

where $T$ denotes temperature estimates subsampled from the original dataset every 1000 years. For better visualization of longer term trends, we smoothed the temperature curve using the loess function in R (stats package) with a span of 0.05.

## (f) Species distribution modelling

Climate data for the last 120 000 years were taken from [53] and converted to a raster stack with resolution $0.5 \times 0.5$ using the R package *raster* [54]. We modelled the climate niche of the pied flycatcher breeding range using four variables: precipitation variability (mm), minimum precipitation (mm), mean temperature (°C) and temperature variability (°C). All predictors showed low to moderate correlation with one another (electronic supplementary material, figure S3). Pied flycatchers are migratory and spend the winter in Africa. As we are interested in the evolutionary history of the species in its breeding grounds, all climate variables represent averages across the breeding season in Europe (May, June and July).

We used two sources of pied flycatcher occurrence data, the 'breeding evidence' dataset collected for the European Breeding Bird Atlas 2 (available from: https://www.ebba2.info/data-availability [24 July 2021]) and GBIF data (available from: https://www.gbif.org [24 July 2021]) for the part of the breeding range east of the Ural mountains. GBIF occurrence data were filtered to include only data collected in May, June or July. The two occurrence datasets were merged and filtered for duplicates. The final occurrence dataset consisted of $N = 2085$ independent records, with one record per $50 \times 50 \text{ km}^2$ cell.

Models were calibrated on current species distribution. Since reliable absence records are not available for our species, we generated background data using biomod2's 'random' approach to sample background points ($N = 6585$) in an area encompassing Europe (with Russia) and Kazakhstan (electronic supplementary material, figure S4). Projections were cross-evaluated by randomly re-sampling data into a calibration (70%) and a validation set (30%). This process was repeated twice. Final individual models were trained with the entire occurrence dataset, and ensembles were built from individual models by taking the mathematical median across predictions with a minimum true skill statistic (TSS; [55]) value $\geq 0.8$.

To identify geographic areas of variable extrapolation, we generated Multivariate Environmental Similarity Surfaces (MESS) using the *mess* function in the dismo R package. In the context of SDMs, MESS calculates the similarity of a given point to the occurrence records for each climate variable, with values less than zero indicating locations where at least one variable was extrapolated [56].

# 3. Results

Variant calling in ANGSD (-doMaf 2) yielded 79 918 sites with a MAF significantly different from zero at a *p*-value < 0.000001.

## (a) Population structure

PCA identified three genetic clusters: a distinct cluster consisting of all six individuals sampled in central Spain ('E', brown), a distinct cluster consisting of 12 of the 15 individuals

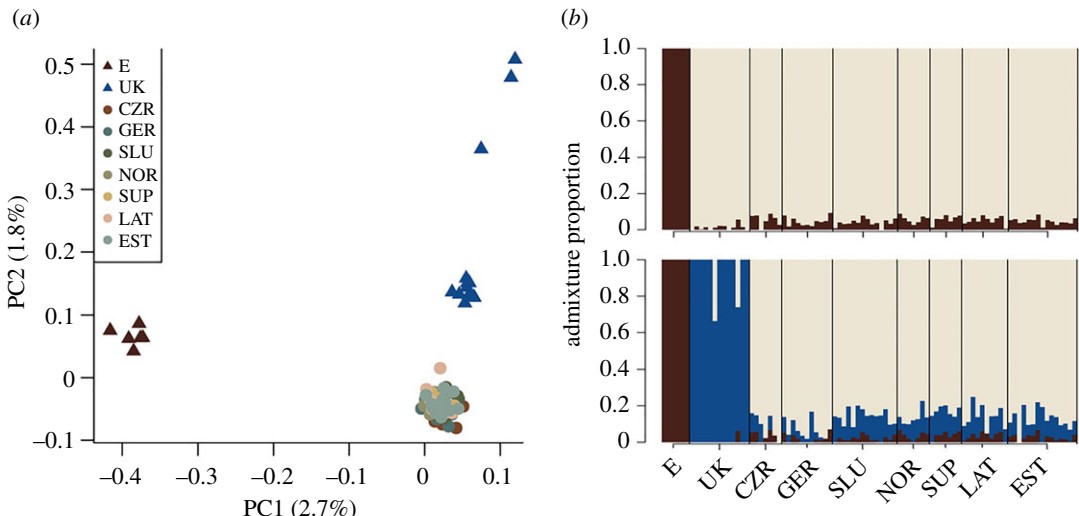

**Figure 1.** Genetic structure of western European pied flycatchers. (a) PCA of 35 686 variable sites. (b) Admixture analysis based on the same set of variable sites. Each individual is represented by a vertical bar, with the colour composition of the bar indicating membership in different genetic clusters. The proportion of cluster membership is indicated on the y-axis. From top to bottom: K = 2, K = 3. (Online version in colour.)

sampled in southern England ('UK', blue) and a third cluster including birds from all other sampling locations (figure 1). The latter will henceforth be referred to as the central and northern European (CNE) cluster. The three clusters in figure 1 are stable over a broad range of MAF thresholds tested (electronic supplementary material, figure S5). At a MAF threshold of 0.05 (35 686 variable sites), the first two PC axes account for 4.4% of the total genetic variation (figure 1).

Admixture analysis using NGSadmix broadly confirms the genetic structure suggested by PCA and previous work based on microsatellite data [31,32]: at all K values > 1, individuals sampled in Spain form a distinct cluster of genetically similar individuals with no evidence of admixture from other European pied flycatchers. At K ≥ 3, all except three individuals sampled in the UK form a distinct genetic cluster; at K ≥ 5, the three UK individuals that appeared as outliers in the PCA (figure 1a) are identified as genetically distinct from the rest of the birds sampled in the UK. At K = 7, both Germany and the Czech Republic form distinct cluster; however, as K values increase beyond K = 7, these become unstable (electronic supplementary material, figure S2).

### (b) Demographic inference

The divergence of all three major European pied flycatcher lineages—Spanish, UK and CNE—coincided with episodes of large-amplitude, rapid climatic change during major glacial–interglacial transitions (figure 2). Although the large CNE cluster is only weakly genetically structured (figure 1; electronic supplementary material, figure S2), we were interested to assess how slight differences in the SFS estimated from different populations within the CNE cluster would affect divergence time estimates. In particular, at some K values, the Czech and German samples each formed more distinct clusters than samples from Scandinavia and the Baltics, respectively (electronic supplementary material, figure S2). To account for these differences, we used three sub-populations for divergence time estimation involving the CNE cluster, German (GER), Czech (CZR) and Latvian (LAT). The different sub-populations of the CNE cluster produced

overlapping divergence time estimates for all combinations except for the Spain–GER combination (figure 2a).

The Spanish lineage diverged from the CNE cluster between 130–102 kya, depending on the sub-population used to represent the CNE cluster (figure 2a, blue lines; electronic supplementary material, figure S6). The Czech and Latvian sub-populations yielded closely overlapping median (inter-quartile range) divergence times of 101.7 (96.9–106.2) kya and 104.4 (101.2–107.5) kya, respectively, consistent with a Spain–CNE split following the end of the Eemian period approximately 115–110 kya (e.g. [57]). However, with the German sub-population, this split was dated to 130 (126.1–134.9) kya, i.e. during the transition into, rather than out of the Eemian (figure 2a,b).

The divergence of the UK lineage from the CNE cluster occurred during the Last Termination, the transition between the last glacial and the current interglacial period *ca* 20–10 kya (figure 2a, orange lines; electronic supplementary material, figure S6). For this more recent split, different sub-populations of CNE (CZR and GER) produced closely overlapping divergence time estimates of 17.7 (16.0–18.9) kya (GER) and 19.8 (18.5–21.5) (CZR), respectively (figure 2a).

### (c) Species distribution modelling

The SDMs suggested that European pied flycatchers experienced fluctuations of suitable climate niche space throughout the late Quaternary (electronic supplementary material, figure S7). However, the only substantial, Europe-wide breeding range collapse occurred 100–104 kya, during the Eemian–Weichselian transition (figure 3; electronic supplementary material, figure S7). At higher temporal resolution, this range collapse appears to be following a succession of rapid climate transitions between 112 and 104 kya, which involved particularly large temperature shifts of up to 16°C [58] (figure 4, inset). The timing of the Spain–CNE split during this time suggests that instability of climate and vegetation at the end of the Eemian led to a population collapse in pied flycatchers that ultimately drove the divergence between the Spanish and CNE lineage.

There is no evidence for an obvious range collapse in pied flycatchers during the entire last glacial period (electronic

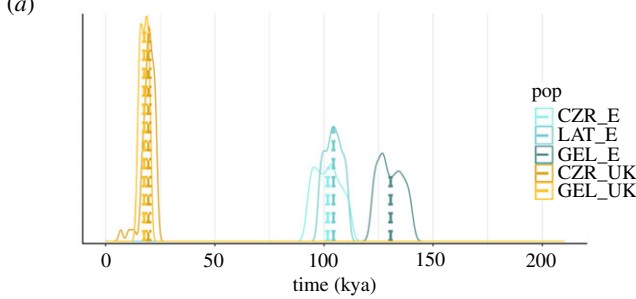

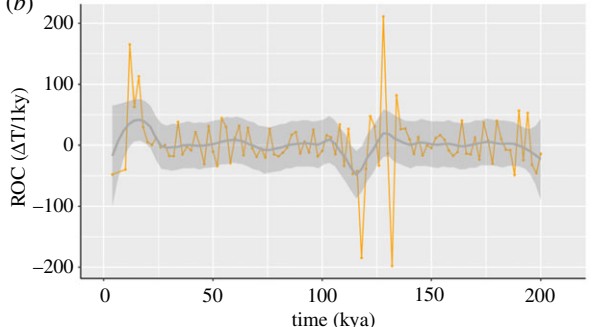

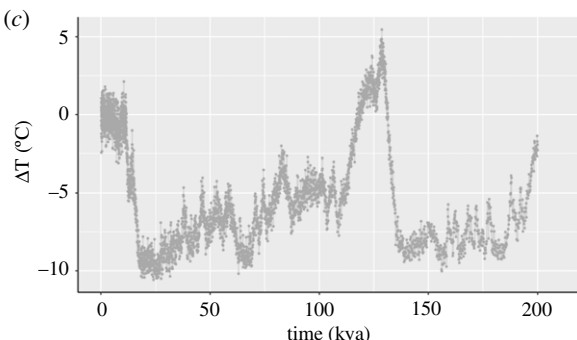

**Figure 2.** Divergence time estimates for the major European pied flycatcher lineages coincide with episodes of climatic instability and high rates of global climate change. (*a*) Time estimates for the divergence between UK–CNE lineages (orange density curves) and Spanish–CNE lineages (blue density curves) for different sub-populations of the CNE cluster. Density curves show the median quartiles (25–75%) of split times (ts). Medians are indicated as vertical lines. (*b*) Rate of climate change per millennium [ΔT/ 1 ky] calculated from temperature data in (*a*) (orange line); (*c*) Global temperatures over the past 200 kya expressed as the difference from the average of the last 1000 years [ΔT]. (Online version in colour.)

supplementary material, figure S7). In particular, suitable climate niche space for pied flycatchers appears to have been widespread throughout Europe even at the height of the LGM, mirroring the distribution of many tree species [60–62] (figure 4*b*; electronic supplementary material, figure S7).

## 4. Discussion

Here, we show that all major European pied flycatcher lineages originated during periods of rapid climate change. Using a direct estimate of the collared flycatcher germ-line mutation rate, we infer divergence times that paint a complex picture of two successive divergence events giving rise to the Spanish lineage, one at the beginning (approx. 130 kya) and one at the end (approx. 110 kya) of the Eemian interglacial period. Based on the traditional view of a near-absence of forests in much of LGM Europe [55], the LGM was thought to have been the primary climate episode shaping current

patterns of genetic diversity in the forest-dependent pied flycatcher [24]. By showing that Spanish pied flycatchers originated during the Eemian rather than during the LGM (26.5–19 kya), we revise the divergence time for this lineage back by more than 100 000 years.

Accepting the assumption made by SDMs that climate niches are stable over time [56], our results call into question the assumption of a post-glacial origin of present-day European pied flycatcher populations in an Iberian refugium. Instead, our niche models imply that much of the Iberian Peninsula has been sub-optimal for pied flycatchers throughout the late Quaternary suggesting that any existing Iberian refugial populations were likely both small and genetically isolated. Finally, the predicted continued availability of suitable pied flycatcher habitat in large areas of Europe throughout the late Pleistocene period points to the region south of the LGM ice sheets as a more plausible source for pied flycatchers currently breeding in CNE.

### (a) Population splits correlate with periods of climate instability, not glacial maxima

Our demographic model suggests that the divergence of both the Spanish and the UK lineages coincides with periods of particularly rapid climate change. Notably, all divergence times inferred here have remarkably narrow interquartile ranges, showing an accuracy of within a few thousand, rather than the more typically obtained tens to hundreds of thousands of years, highlighting the potential of direct mutation rates for molecular dating of intra-specific events.

We obtained split time estimates of 130 kya with the German sub-population representing the CNE cluster and 104 and 108 kya, respectively, with two CNE sub-populations east (Czech Republic) and north (Latvia) of Germany. Based on these observations, two scenarios can be envisaged: I) two successive divergence events, one during the transition into and one during the transition out of the Eemian. Or II) a protracted divergence process that began during the transition into the Eemian and was reinforced by the second phase of climate instability during the transition into the following glacial period (Weichselian) [58,63,64]. Given the uncertainty in absolute age estimates from both ice core data (up to 6 kya, [65]) and genetic data (approx. 20%, [49]), it is remarkable how clearly each divergence event involving the Spanish lineage correlates with a different glacial–interglacial transition period. Based on this, we are inclined to favour the scenario of two successive divergence events, one prompted by climate instability during the transition into the Eemian, and one following successive high-amplitude and rapid oscillations characterizing the transition between the Eemian and the Weichselian between 115 and 108 kya.

The divergence of the UK lineage from the large CNE group occurred between 20 and 18 kya during the transition between the end of the last cold stage and the start of the Holocene 20–10 kya. Overlapping interquartile ranges around divergence dates from models with different CNE subgroups suggest a single event during which the UK lineage diverged from the common ancestor of the weakly structured CNE cluster. Ice core data and palaeobotanical records indicate that the last deglaciation was punctuated by climatic oscillations on timescales of a few thousand years [66]. The time preceding the UK split was marked by a series of stadial–interstadial transitions (GI-2.1 (23 020) to

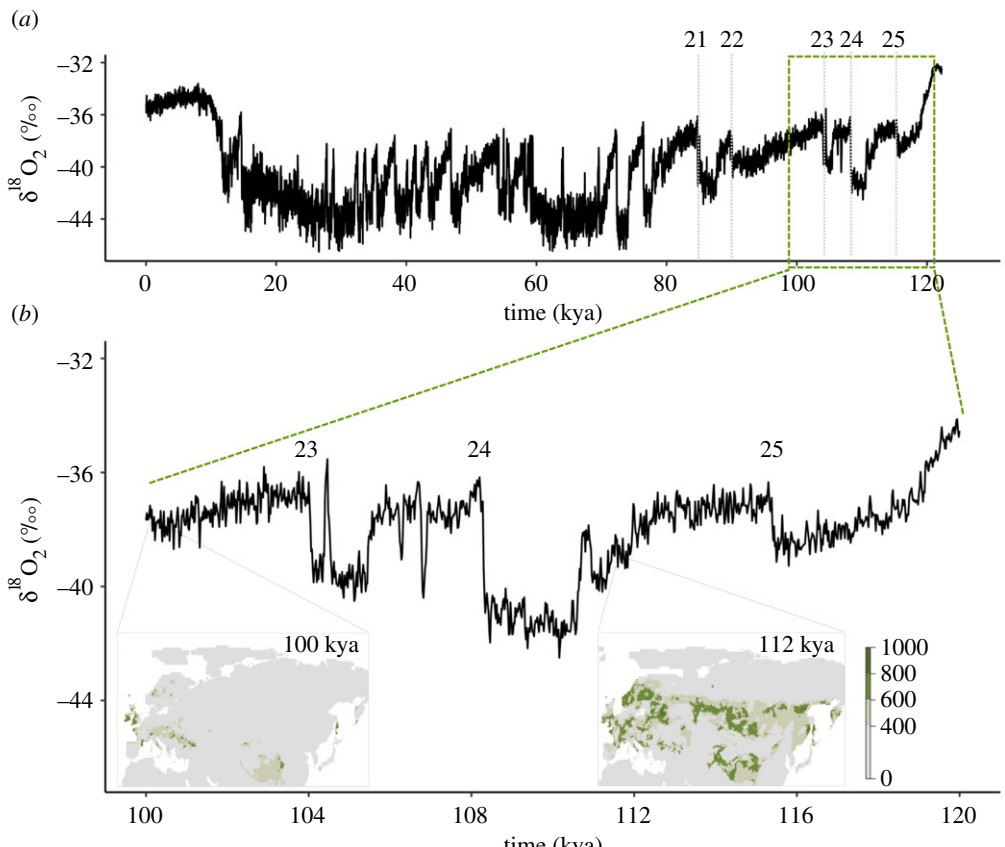

**Figure 3.** (a) Chronology of climatic events over the last 120 000 years visible in the dO18 record of the NGRIP ice core (black line) and approximate position of transitions between Greenland Stadials (GS) 25–21 and associated interstadials (grey dotted lines). Labelling of events after (87). (b) The projected range collapse of pied flycatchers around 100 kya (insets) follows a series of rapid changes in average global temperature. (Online version in colour.)

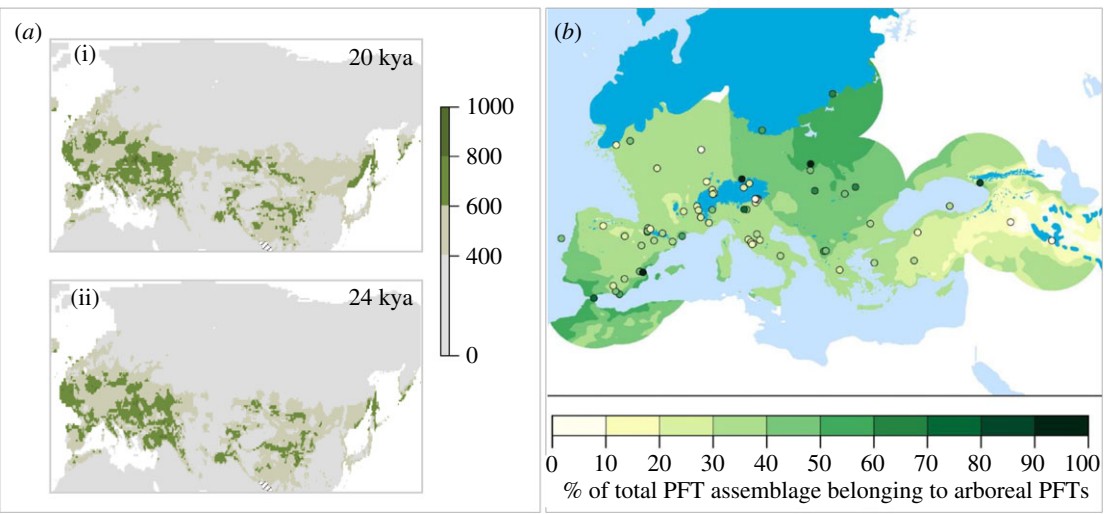

**Figure 4.** (a) Climate niche projections for pied flycatchers for the early (i) and late (ii) LGM, respectively. Hatched areas denote MESS values lower than −5000 for at least one variable. For the full set of MESS maps, see electronic supplementary material, figure S8. (b) Pollen-based tree cover reconstruction for the late LGM (22–19 kya). Tree cover reconstruction adapted from Kaplan et al. [59]. PFT, plant functional types. (Online version in colour.)

GS-2.1c (22 900), b (20 900) and c (17 480), respectively, [67]). However, the most prominent climatic changes following the end of the LGM occurred at around 14 kya (Bølling warm phase) and 13 kya (Younger Dryas cooling event), respectively, i.e. after the divergence of the UK lineage. The absence of any clear link with major climatic events prevents us from establishing climate change as the dominant driver of the UK split, although indirect effects, for example via different refugia for deciduous versus coniferous trees, might be possible. SDM predict higher coniferous than deciduous species diversity for the UK regions during the LGM [68]. Although the modern-day habitat preference of UK breeding pied flycatchers is almost exclusively deciduous forest, they do breed in riparian mixed coniferous/deciduous forest, and so could have persisted in a coniferous dominated landscape.

## (b) Population divergence follows widespread climate-induced habitat contractions

The limited availability of high-resolution palaeoclimate data for pre-LGM Eurasia [53] has meant that little is known about how climate conditions other than those prevailing during the LGM and the Holocene period have affected biota. Glacial–interglacial transitions were characterized by a frequent occurrence of short warming events and jumps in climate conditions [58,69]. To investigate the impact of such episodes of pronounced climate instability on the availability and extent of suitable pied flycatcher habitat, we took advantage of recently published high-resolution palaeoclimate data dating back 120 kya to model this species' climate niche space through time.

Our SDMs suggest a widespread collapse in suitable climate niche space for pied flycatchers 104–100 kya, i.e. towards the end of the Eemian–Weichselian transition. The timing of this range collapse closely follows the first marked cooling over Greenland 110–108 kya (Greenland Stadial (GS) 25), which ended with a particularly extreme temperature increase of 16°C [70]. Based on pollen and plant macrofossils, the vegetation in Europe started to change towards open, tundra-like habitat during this time, beginning around 117.5 kya in northern Europe and reaching southern Europe by around 110 kya [71,72]. The breeding success of pied flycatchers is highly dependent on tree habitat and tends to be higher in deciduous versus coniferous forest [73–75]. Our climate niche models predict a widespread breeding range collapse for pied flycatchers around this time, as would be expected for this forest-dependent species. Combining the results from demographic and climate niche modelling, we conclude that the divergence of the Spanish lineage between 102 and 104 kya was driven by climatic and environmental instability during the Eemian–Weichselian transition.

## (c) Assessing the Iberian glacial refugium hypothesis

Based on their current presence in central Spain, but not in any of the other classical southern refugia in Europe (Italy, the Balkans–Greece, the Caspian/Caucasus region, [76]), pied flycatchers are assumed to have survived the LGM in the Iberian Peninsula [24]. Contrary to these assumptions, we found no support for a special role of Iberia as a glacial refugium for this species. Instead, our SDM models suggest that large areas to the south and east of the Northern Hemisphere ice sheets remained climatically suitable for pied flycatchers, even at the height of the LGM. The plausibility of a continued presence of pied flycatchers across Europe is supported by tree pollen records for that time, which suggest that tree cover exceeded 30%, and reached up to 60% in much of LGM Europe [59,62].

The importance of Iberia for the glacial survival of the species is further questioned by the recent observation that the historical effective population size ($N_e$) of Spanish pied flycatchers is substantially smaller than that of other European populations, including that of an island population [77]. Under the proposed scenario of a post-glacial expansion from an Iberian source population, the opposite would be expected. Similarly, under the Iberian refugium hypothesis, we would expect the Spanish population to have higher genetic variation and lower inbreeding than populations in areas that were supposedly re-colonized from an Iberian source. However, again, the opposite is observed. Based on these findings, Kardos

*et al.* [77] suggested two alternative scenarios: (i) the Spanish population was not the source population for other European populations; (ii) demographic events affecting only the Spanish population caused the observed pattern of low genetic variation and high inbreeding.

Our predictions of limited habitat availability in the Iberian Peninsula during the LGM, together with predictions of the continued availability of suitable habitat south and east of the ice sheet, would make a re-colonization of northern Europe from the latter area seem much more likely than a re-colonization from Iberia. Finally, if pied flycatchers did persist in the Iberian Peninsula during the LGM, population sizes were probably never large, consistent with genetic signatures of historically low $N_e$, low genetic variation and high inbreeding in Spanish pied flycatchers [77]. A historically low $N_e$ has recently also been suggested for the Iberian refugial population of dunnocks, *Prunella modularis*, similarly suggesting that the Iberian dunnock lineage played no role in the recolonization of Holocene Europe [27].

Whereas the Iberian Peninsula does not stand out as a major glacial refugium for pied flycatchers, our niche models predict their presence during this time in Italy and the Balkan Peninsula (electronic supplementary material, figure S6). Another area projected to have been suitable for pied flycatchers during the LGM is northern Africa (electronic supplementary material, figure S7). Their current absence in these, and other classical southern refugia, might reflect true unsuitability or might be due to competition with other, ecologically similar bird species, notably the Atlas flycatcher (*F. speculigera*) in northern Africa. Multi-species SDMs including all four black and white *Ficedula* flycatchers might shed more light on the role of climate versus competition in shaping species distributions in this taxonomic group in future studies.

## 5. Conclusion

Many intra-specific diversification events occur on timescales of a few thousand years [78]. Achieving this fine temporal resolution requires accurate estimates of the rate at which mutations accumulate in the species of interest. Direct mutation rate estimates offer the possibility for more accurate dating than is typically achieved using mutation rates calculated indirectly from dated phylogenies. Using the first direct mutation rate for passerine birds, we find clear evidence for rapid global climatic change as an important driver of population divergence in a forest-dependent bird species. By unambiguously linking all major divergence events in pied flycatchers to episodes of rapid global climate change, our results highlight the need to better understand the impact on the natural world of climate change dimensions other than those prevailing during the last glacial maximum.

Ethics. Molecular data presented in this paper were generated from existing DNA extracts (details in Lehtonen *et al.* 2009, Mol Ecol [31]).

Data accessibility. All data required to reproduce our results are available from the Dryad Digital Repository: https://doi.org/10.5061/dryad.stqjq2c3x [79].

Authors' contributions. V.M.W.: conceptualization, data curation, formal analysis, investigation, methodology, project administration, validation, visualization, writing-original draft, writing-review and editing; M.D.B.: resources, writing-review and editing; M.M.: resources, writing-review and editing; T.L.: resources, writing-review and editing; A.M.: resources, writing-review and editing; A.N.: resources, writing-review and editing; C.R.P.: resources, writing-review and editing; G.S.: resources, writing-review and

editing; W.W.: resources, writing-review and editing; H.E.: funding acquisition, project administration, resources, supervision, writing-review and editing. All authors gave final approval for publication and agreed to be held accountable for the work performed therein.

**Competing interests.** We declare we have no competing interests.

**Funding.** This work was primarily supported by grants from the Knut and Alice Wallenberg foundation (grant no. 2014/0044) and the Swedish Research Council (grant no. 2013-8271) to H.E. A.N. was supported by the Swedish Research Council (grant no. 2020-04686) and the Birgit and Hellmuth Hertz Foundation of The Royal Physiographic Society of Lund (grant no. 2017-39034); C.R.P. by the Academy of Finland (grant no. 314254); M.M. by the Estonian Research Council (grant no. IUT34-8) and T.L. by the Emil Aaltonen Foundation.

**Acknowledgements.** We thank Paula Leskinen for DNA extractions of some of the material included in the study.

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
