## [Peer Review File · Proceedings of the Royal Society B: Biological Sciences]

Review History

RSPB-2021-1066.R0 (Original submission)

Review form: Reviewer 1

Recommendation

Accept with minor revision (please list in comments)

Scientific importance: Is the manuscript an original and important contribution to its field?

Excellent

General interest: Is the paper of sufficient general interest?

Excellent

Quality of the paper: Is the overall quality of the paper suitable?

Excellent

Is the length of the paper justified?

Yes

Should the paper be seen by a specialist statistical reviewer?

No

Do you have any concerns about statistical analyses in this paper? If so, please specify them explicitly in your report.

No

It is a condition of publication that authors make their supporting data, code and materials available - either as supplementary material or hosted in an external repository. Please rate, if applicable, the supporting data on the following criteria.

Is it accessible?

Yes

Is it clear?

N/A

Is it adequate?

N/A

Do you have any ethical concerns with this paper?

No

Comments to the Author

This paper is a welcome addition to the Ficedula Files! It cautions that accurate knowledge of mutation rates can profoundly affect divergence time estimates - that is obvious enough - but that default attribution of within-species divergences to LGM dynamics can be faulty. It is the latter caution that is timely and always worth hearing as are the analyses behind it.

Colours in K= 4 and 5 in Figure 2 are hard to make out. Do you even need to show those values? Figure 3's labelling seems to have gone awry and should definitely be checked and fixed.

Why not refer to the subspecies taxonomy of the populations as per my comment box alongside Figure 1? And how might the African form *F. h. speculigera* fit into this story? Perhaps a sentence for future work in the Discussion could be added.

Review form: Reviewer 2

Recommendation

Accept with minor revision (please list in comments)

Scientific importance: Is the manuscript an original and important contribution to its field?

Good

General interest: Is the paper of sufficient general interest?

Good

Quality of the paper: Is the overall quality of the paper suitable?

Good

Is the length of the paper justified?

Yes

Should the paper be seen by a specialist statistical reviewer?

No

Do you have any concerns about statistical analyses in this paper? If so, please specify them explicitly in your report.

No

It is a condition of publication that authors make their supporting data, code and materials available - either as supplementary material or hosted in an external repository. Please rate, if applicable, the supporting data on the following criteria.

Is it accessible?

Yes

Is it clear?

Yes

Is it adequate?

Yes

Do you have any ethical concerns with this paper?

No

Comments to the Author

Comments to the Author:

This paper makes a case for how rapid climate change can lead to population splits. The divergence time was estimated by DADI using 2D-SFS for each population pairs. I suggest to complementally infer the demographic models by incorporating all 3 major populations in one time instead of pair by pair inferences.

Line 81: I guess you used mutation rate from ref.44 rather than from ref.19 here, as same in your dadi analysis.

Line 159 : Please elaborate further on this model.

Line 248-293: I guess the panels in figure 3 were erroneously arranged.

Line 355: change 26.5 to 26.5 - 19. It would be much better to explain a little bit why is the previous divergence time estimation quite different from the current study. From the current descriptions, the origination of the previous divergence time is obscure.

Figure 2A: As some overlapped circles will lead to darker colors, and it's not easy to discriminate "E" from some of the CNE populations, so it may be better to change circles to other symbols of "E" and "UK".

Review form: Reviewer 3

Recommendation

Major revision is needed (please make suggestions in comments)

Scientific importance: Is the manuscript an original and important contribution to its field?

Excellent

General interest: Is the paper of sufficient general interest?

Good

Quality of the paper: Is the overall quality of the paper suitable?

Good

Is the length of the paper justified?

Yes

Should the paper be seen by a specialist statistical reviewer?

Yes

Do you have any concerns about statistical analyses in this paper? If so, please specify them explicitly in your report.

Yes

It is a condition of publication that authors make their supporting data, code and materials available - either as supplementary material or hosted in an external repository. Please rate, if applicable, the supporting data on the following criteria.

Is it accessible?

Yes

Is it clear?

Yes

Is it adequate?

Yes

Do you have any ethical concerns with this paper?

No

Comments to the Author

Dear authors,

Your manuscript is well written, structured and most importantly it deals with a very interesting topic. My comments mainly focus on the ENM part of your study. Here, I see a few drawbacks that can unfortunately only be tackled by a reanalysis of the data. As the ENM part is important for your study, I believe it can be substantially improved. I missed a more intensive engagement with ENMs (e.g. references, theoretical background, more algorithms, occurrence and background sampling should be improved etc.). I hope that my comments will help to improve your fascinating research.

Kind regards

Detailed comments

Line 52 ff Introduction

Your introduction includes a short paragraph on the progress in molecular methods and you cite articles using molecular methods. However, a paragraph on ENMs is lacking and I missed at least a few standard references for SDMs/ENMs (e.g. Elith et al. 2006, *Ecography*; Guisan & Thuiller 2005, *Ecol. Let.*; Elith & Leathwick 2009, *AREES*; Engler et al 2017, *J. Avian Biol.* for avian SDMs). ENMs were also used for reconstructing the influence of Pleistocene climate fluctuations and this should not be ignored (some are perhaps better included in the discussion). You might refer to Rödder et al. 2013, *PLoSOne* (they even reconstructed ancestral niches); Peterson & Anamza 2016, *J. Avian Biol.*; Schidelko et al. 2011, *Biol J Lin. Soc.*; Schidelko et al 2013, *J. Avian Biol.* (both refer to diversity patterns), Kamp et al. 2018, *Zool. Scripta* (Phylogeography + SDM). Note also that your approach assumes that the climate niche of the Pied Flycatcher is stable over time (check Engler et al. 2017 and the references therein for a brief review on niche conservatism

in birds). As this is a prerequisite for your approach, it should be stated somewhere in a brief sentence.

Line 61ff

Please add a few older references. While many details were discovered during the last 25 years by new methods, the importance of the Pleistocene period for today's biota has been known or hypothesised at least for decades (e.g. see Haffer 1969 and many more).

Line 80

Are these data transferrable from one species to another (perhaps add a half-sentence why this is ok)?

Line 91

Data from the east of the distribution is obviously missing. Given your focus on the Iberian Population this seems to be ok for me, but you should give a half-sentence why your sampling effort is adequate (see also comment on nomenclature in ESM).

Line 188

biomod2 offers different modelling approaches, all of them have different strengths and drawbacks. Why did you focus on these four (why not taking nearly all), e.g. why did you ignore Maxent (perhaps the most prominent of the available algorithms)? I can only assume that computation time is the limiting factor but using only four approaches somehow contradicts the sense of an ensemble approach to me.

Line 190/191

I must have missed it, but while I found model specifications I did not find the model evaluations (e.g. TSS, AUC) in the ESM.

Line 193

Why did you take August instead of May? Most birds should have returned by mid May, and by the end of July the species has largely left its breeding grounds as far as I know? Please correct me if I am wrong as I am most familiar with the situation in Central Europe.

In general, it is very good to see that you limit your ENM to climate from the breeding season (but you should explain to non-ornithologists that the species is migratory and not present in the area during other months in a half-sentence).

Line 200ff

From my point of view, using occurrence data from a distribution map in a single species approach is far from current standards (you mention the need for improvement in Spain but these problems could occur anywhere). I am aware that it takes some work but for a single species, actual observations should be used, e.g. from GBIF and observer platforms (ornitho, observation.org; eBird shares its data with GBIF etc.). This makes modelling more complicated (observer bias etc.) but these aspects could be tackled by appropriate background sampling etc. Using occurrences from Birdlife Polygons might be adequate for a macroecological study with a lot of species but not for a single species. Also note the fine scale climate data you use and bird and climate data should fit together. I understand that your approach might also be necessary east of the Ural where data are sparse (check Fourcade et al. 2013, Biol Conserv for a possible to solution to different sampling effort across Eurasia) but high-quality data from Europe should be available (see also EBBA 2). Thinning of the occurrence data was good but did not tackle the mentioned problems.

Background sampling (as detailed in the ESM) is also unusual and not adequate for your research questions. While a migratory species might reach a larger area than a small let's say amphibian species your background is far too large (overfitting). Better use a buffer for background sampling or sample background points from ecoregions where the species is recorded (check e.g. Guisan et al. 2014, Trend. Ecol Evol; Stiel et al. 2021, JOrn for examples). Better take 10.000 background points, and check if background points are taken in a similar way than your

occurrence data (degree grid cells are larger close to the pole and smaller in temperate regions – given the latitude your data span, this should not be ignored). I think, you could use the *dismo* R package for this purpose.

Please highlight areas of model uncertainty. Typically, your models extend (in different ways) beyond the training climate. It is necessary to show novel climates. You should make MESS maps (areas could be given as hatched overlays in the maps) (Elith et al. 2010, *Methods Ecol Evol*).

Line 215

In the ESM you wrote that the TSS value was set to 0.8 (see below). Please check.

Line 390 ff

The figure 5 on tree cover was a very interesting bonus. Reading your discussion, it would be interesting to know a bit more about deciduous vs. coniferous forests during the LGM and later on. You discuss this issue for earlier times in line 418 ff for the split of the Spanish population. However, as far as I know the British population avoids nearly any coniferous forests. In the discussion (line 390ff), you dismiss a direct link to climate fluctuations. However, indirect effects via different refugia for deciduous and coniferous forests seem to be possible? Please correct me if I am wrong. If you agree, please include a sentence on deciduous vs coniferous forest preferences for the split of the British population.

Line 405

“little is known” - But see references using SDMs mentioned above.

Line 438

Consider a brief sentence on biotic interactions with other flycatchers (e.g. Semicollared in the Caucasus which is present as refugium in your ENM). Could this have affected the observed biogeographical pattern? It is possible that the fundamental niche is underestimated your model (Soberón & Peterson 2005, *Biodivers. Informatics for niche concept*) – the Pied Flycatcher might simply avoid the Caucasus due to competition with its congener. As far as I understood from one of your articles (Sætre & Sæther 2010, *Mol. Ecol.*) competition could (at least theoretically) alter the biogeography of the Pied Flycatcher as well. Putative refugia in the Maghreb and the Caucasus are inhabited by Atlas FC and Semicollared FC. And the absence of the Pied FC today (or in the past?) in these areas might not be explained by climate factors but by competition. This is to some degree speculative but a proper model projected onto these areas can give insight into this question. Otherwise, your results must be interpreted much more carefully. This is another reason why you should definitely use a smaller background area and subsequently project your model onto the larger area shown in your maps.

ESM

Line ESM-36

Great to see an abstract on nomenclature. I was already wondering why *F. speculigera* is not included in your approach but as a species on its own (basal to Collared and Pied) it makes sense to omit occurrence data within the current framework of your study.

Please, could you add a short sentence on *F. h. sibirica*? Your genetic sampling does not cover this subspecies but your ENM covers its distribution (which makes sense). Even if you do not want to speculate on e.g. refugia for this taxon, a brief mention seems adequate (e.g. line 91).

Line ESM-228

You wrote in the manuscript, that TSS threshold was set to 0.7. (Line 215) Which value is correct?

Decision letter (RSPB-2021-1066.R0)

09-Jun-2021

Dear Dr Warmuth:

Your manuscript has now been peer reviewed and the reviews have been assessed by an Associate Editor. The reviewers' comments (not including confidential comments to the Editor) and the comments from the Associate Editor are included at the end of this email for your reference. As you will see, the reviewers and the Editors have raised some concerns with your manuscript and we would like to invite you to revise your manuscript to address them.

Research ethics:

Use of animals and field studies:

It is a condition of publication that you make available the data and research materials supporting the results in the article. Please see our Data Sharing Policies (<https://royalsociety.org/journals/authors/author-guidelines/#data>). Datasets should be deposited in an appropriate publicly available repository and details of the associated accession number, link or DOI to the datasets must be included in the Data Accessibility section of the article (<https://royalsociety.org/journals/ethics-policies/data-sharing-mining/>). Reference(s) to datasets should also be included in the reference list of the article with DOIs (where available).

If you wish to submit your data to Dryad (<http://datadryad.org/>) and have not already done so you can submit your data via this link [http://datadryad.org/submit?journalID=RSPB&manu=\(Document not available\)](http://datadryad.org/submit?journalID=RSPB&manu=(Document%20not%20available)), which will take you to your unique entry in the Dryad repository.

Please submit a copy of your revised paper within three weeks. If we do not hear from you within this time your manuscript will be rejected. If you are unable to meet this deadline please let us know as soon as possible, as we may be able to grant a short extension.

Best wishes,
Dr Locke Rowe
mailto: proceedingsb@royalsociety.org

Associate Editor
Board Member: 1
Comments to Author:

Dear authors,

I have now received 3 reviewer reports and have read the MS myself. As you will see from their comments whilst everyone thought the topic of this paper is interesting, they also raised a number of concerns, which I share. Reviewer 2 makes some useful suggestions for reanalysis of divergence times by incorporating all 3 major populations in one time instead of pair by pair inferences. Referee 3 has concerns about the environmental niche modelling that need to be addressed. Below, the reviewers provide some additional constructive suggestions for the authors to consider. Comments from reviewer 1 can be found in the attached pdf.

Reviewer(s)' Comments to Author:
Referee: 1

Comments to the Author(s)

This paper is a welcome addition to the Ficedula Files! It cautions that accurate knowledge of mutation rates can profoundly affect divergence time estimates - that is obvious enough - but that default attribution of within-species divergences to LGM dynamics can be faulty. It is the latter caution that is timely and always worth hearing as are the analyses behind it.

Colours in K= 4 and 5 in Figure 2 are hard to make out. Do you even need to show those values? Figure 3's labelling seems to have gone awry and should definitely be checked and fixed.

Why not refer to the subspecies taxonomy of the populations as per my comment box alongside Figure 1? And how might the African form *F. h. speculigera* fit into this story? Perhaps a sentence for future work in the Discussion could be added.

Referee: 2

Comments to the Author(s)

Comments to the Author:

This paper makes a case for how rapid climate change can lead to population splits. The divergence time was estimated by DADI using 2D-SFS for each population pairs. I suggest to complementally infer the demographic models by incorporating all 3 major populations in one time instead of pair by pair inferences.

Line 81: I guess you used mutation rate from ref.44 rather than from ref.19 here, as same in your dadi analysis.

Line 159 : Please elaborate further on this model.

Line 248-293: I guess the panels in figure 3 were erroneously arranged.

Line 355: change 26.5 to 26.5 – 19. It would be much better to explain a little bit why is the previous divergence time estimation quite different from the current study. From the current descriptions, the origination of the previous divergence time is obscure.

Figure 2A: As some overlapped circles will lead to darker colors, and it's not easy to discriminate "E" from some of the CNE populations, so it may be better to change circles to other symbols of "E" and "UK".

Referee: 3

Comments to the Author(s)

Dear authors,

Your manuscript is well written, structured and most importantly it deals with a very interesting topic. My comments mainly focus on the ENM part of your study. Here, I see a few drawbacks that can unfortunately only be tackled by a reanalysis of the data. As the ENM part is important for your study, I believe it can be substantially improved. I missed a more intensive engagement with ENMs (e.g. references, theoretical background, more algorithms, occurrence and background sampling should be improved etc.). I hope that my comments will help to improve your fascinating research.

Kind regards

Detailed comments

Line 52 ff Introduction

Your introduction includes a short paragraph on the progress in molecular methods and you cite articles using molecular methods. However, a paragraph on ENMs is lacking and I missed at least a few standard references for SDMs/ENMs (e.g. Elith et al. 2006, *Ecography*; Guisan & Thuiller 2005, *Ecol. Let.*; Elith & Leathwick 2009, *AREES*; Engler et al 2017, *J. Avian Biol.* for avian SDMs). ENMs were also used for reconstructing the influence of Pleistocene climate fluctuations and this should not be ignored (some are perhaps better included in the discussion). You might

refer to Rödder et al. 2013, PLoSOne (they even reconstructed ancestral niches); Peterson & Anamza 2016, J. Avian Biol.; Schidelko et al. 2011, Biol J Lin. Soc.; Schidelko et al 2013, J. Avian Biol. (both refer to diversity patterns), Kamp et al. 2018, Zool. Scripta (Phylogeography + SDM). Note also that your approach assumes that the climate niche of the Pied Flycatcher is stable over time (check Engler et al. 2017 and the references therein for a brief review on niche conservatism in birds). As this is a prerequisite for your approach, it should be stated somewhere in a brief sentence.

Line 61ff

Please add a few older references. While many details were discovered during the last 25 years by new methods, the importance of the Pleistocene period for today's biota has been known or hypothesised at least for decades (e.g. see Haffer 1969 and many more).

Line 80

Are these data transferrable from one species to another (perhaps add a half-sentence why this is ok)?

Line 91

Data from the east of the distribution is obviously missing. Given your focus on the Iberian Population this seems to be ok for me, but you should give a half-sentence why your sampling effort is adequate (see also comment on nomenclature in ESM).

Line 188

biomod2 offers different modelling approaches, all of them have different strengths and drawbacks. Why did you focus on these four (why not taking nearly all), e.g. why did you ignore Maxent (perhaps the most prominent of the available algorithms)? I can only assume that computation time is the limiting factor but using only four approaches somehow contradicts the sense of an ensemble approach to me.

Line 190/191

I must have missed it, but while I found model specifications I did not find the model evaluations (e.g. TSS, AUC) in the ESM.

Line 193

Why did you take August instead of May? Most birds should have returned by mid may, and by the end of July the species has largely left its breeding grounds as far as I know? Please correct me if I am wrong as I am most familiar with the situation in Central Europe.

In general, it is very good to see that you limit your ENM to climate from the breeding season (but you should explain to non-ornithologists that the species is migratory and not present in the area during other months in a half-sentence).

Line 200ff

From my point of view, using occurrence data from a distribution map in a single species approach is far from current standards (you mention the need for improvement in Spain but these problems could occur anywhere). I am aware that it takes some work but for a single species, actual observations should be used, e.g. from GBIF and observer platforms (ornitho, observation.org; eBird shares its data with GBIF etc.). This makes modelling more complicated (observer bias etc.) but these aspects could be tackled by appropriate background sampling etc. Using occurrences from Birdlife Polygons might be adequate for a macroecological study with a lot of species but not for a single species. Also note the fine scale climate data you use and bird and climate data should fit together. I understand that your approach might also be necessary east of the Ural where data are sparse (check Fourcade et al. 2013, Biol Conserv for a possible to solution to different sampling effort across Eurasia) but high-quality data from Europe should be available (see also EBBA 2). Thinning of the occurrence data was good but did not tackle the mentioned problems.

Background sampling (as detailed in the ESM) is also unusual and not adequate for your research questions. While a migratory species might reach a larger area than a small let's say amphibian species your background is far too large (overfitting). Better use a buffer for background sampling or sample background points from ecoregions where the species is recorded (check e.g. Guisan et al. 2014, *Trend. Ecol Evol*; Stiels et al. 2021, *JOrn* for examples). Better take 10.000 background points, and check if background points are taken in a similar way than your occurrence data (degree grid cells are larger close to the pole and smaller in temperate regions - given the latitude your data span, this should not be ignored). I think, you could use the *dismo* R package for this purpose.

Please highlight areas of model uncertainty. Typically, your models extend (in different ways) beyond the training climate. It is necessary to show novel climates. You should make MESS maps (areas could be given as hatched overlays in the maps) (Elith et al. 2010, *Methods Ecol Evol*).

Line 215

In the ESM you wrote that the TSS value was set to 0.8 (see below). Please check.

Line 390 ff

The figure 5 on tree cover was a very interesting bonus. Reading your discussion, it would be interesting to know a bit more about deciduous vs. coniferous forests during the LGM and later on. You discuss this issue for earlier times in line 418 ff for the split of the Spanish population. However, as far as I know the British population avoids nearly any coniferous forests. In the discussion (line 390ff), you dismiss a direct link to climate fluctuations. However, indirect effects via different refugia for deciduous and coniferous forests seem to be possible? Please correct me if I am wrong. If you agree, please include a sentence on deciduous vs coniferous forest preferences for the split of the British population.

Line 405

"little is known" - But see references using SDMs mentioned above.

Line 438

Consider a brief sentence on biotic interactions with other flycatchers (e.g. Semicollared in the Caucasus which is present as refugium in your ENM). Could this have affected the observed biogeographical pattern? It is possible that the fundamental niche is underestimated your model (Soberón & Peterson 2005, *Biodivers. Informatics for niche concept*) - the Pied Flycatcher might simply avoid the Caucasus due to competition with its congener. As far as I understood from one of your articles (Sætre & Sæther 2010, *Mol. Ecol.*) competition could (at least theoretically) alter the biogeography of the Pied Flycatcher as well. Putative refugia in the Maghreb and the Caucasus are inhabited by Atlas FC and Semicollared FC. And the absence of the Pied FC today (or in the past?) in these areas might not be explained by climate factors but by competition. This is to some degree speculative but a proper model projected onto these areas can give insight into this question. Otherwise, your results must be interpreted much more carefully. This is another reason why you should definitely use a smaller background area and subsequently project your model onto the larger area shown in your maps.

ESM

Line ESM-36

Great to see an abstract on nomenclature. I was already wondering why *F. speculigera* is not included in your approach but as a species on its own (basal to Collared and Pied) it makes sense to omit occurrence data within the current framework of your study.

Please, could you add a short sentence on *F. h. sibirica*? Your genetic sampling does not cover this subspecies but your ENM covers its distribution (which makes sense). Even if you do not want to speculate on e.g. refugia for this taxon, a brief mention seems adequate (e.g. line 91).

Line ESM-228

You wrote in the manuscript, that TSS threshold was set to 0.7. (Line 215) Which value is correct?

Author's Response to Decision Letter for (RSPB-2021-1066.R0)

See Appendix A.

RSPB-2021-1066.R1 (Revision)

Review form: Reviewer 2

Recommendation

Accept as is

Scientific importance: Is the manuscript an original and important contribution to its field?

Good

General interest: Is the paper of sufficient general interest?

Good

Quality of the paper: Is the overall quality of the paper suitable?

Good

Is the length of the paper justified?

Yes

Should the paper be seen by a specialist statistical reviewer?

No

Do you have any concerns about statistical analyses in this paper? If so, please specify them explicitly in your report.

No

It is a condition of publication that authors make their supporting data, code and materials available - either as supplementary material or hosted in an external repository. Please rate, if applicable, the supporting data on the following criteria.

Is it accessible?

Yes

Is it clear?

Yes

Is it adequate?

Yes

Do you have any ethical concerns with this paper?

No

Comments to the Author

Comments to the Authors:

I think the authors have made a good job to revise the ms, both in terms of the new analyses and in rewriting, and thus I have no more comments to add.

Review form: Reviewer 3

Recommendation

Accept with minor revision (please list in comments)

Scientific importance: Is the manuscript an original and important contribution to its field?

Excellent

General interest: Is the paper of sufficient general interest?

Excellent

Quality of the paper: Is the overall quality of the paper suitable?

Excellent

Is the length of the paper justified?

Yes

Should the paper be seen by a specialist statistical reviewer?

No

Do you have any concerns about statistical analyses in this paper? If so, please specify them explicitly in your report.

No

It is a condition of publication that authors make their supporting data, code and materials available - either as supplementary material or hosted in an external repository. Please rate, if applicable, the supporting data on the following criteria.

Is it accessible?

Yes

Is it clear?

Yes

Is it adequate?

Yes

Do you have any ethical concerns with this paper?

No

Comments to the Author

Dear authors,

thank you for following the recommendations. I believe that your manuscript has been substantially improved now and is a really fascinating paper. I am absolutely content with your changes and only have one minor concern left: The raw output of the Mess maps is very difficult to interpret. I would recommend showing areas of non-analogous climate hatched (e.g. in figure 4a). Areas of model uncertainty should be immediately visible to the reader, e.g. you might include them as hatched area in figure 4A.

Two other issues should be checked for grammar:

Line 616 – „Accepting the assumption...“ Thanks for including this recommend content but I believe somehow this phrase is not a complete English sentence? Please check grammar.

Line 544ff – Please check scientific species names in your references - they should be written in italics. I might have missed a few names but at least check also line 538 and 571.

Kind regards

Decision letter (RSPB-2021-1066.R1)

01-Oct-2021

Dear Dr Warmuth

I am pleased to inform you that your manuscript RSPB-2021-1066.R1 entitled "Major population splits coincide with episodes of rapid climate change in a forest-dependent bird" has been accepted for publication in Proceedings B.

The referee(s) have recommended publication, but also suggest some minor revisions to your manuscript. Therefore, I invite you to respond to the referee(s)' comments and revise your manuscript. Because the schedule for publication is very tight, it is a condition of publication that you submit the revised version of your manuscript within 7 days. If you do not think you will be able to meet this date please let us know.

Sincerely,

Dr Locke Rowe

Associate Editor:

Comments to Author:

Dear authors,

the manuscript has been reviewed by two of the previous reviewers who were happy with the revisions made. Referee 2 only has a few minor suggestions considering the figures and a few grammatical suggestions.

Reviewer(s)¹ Comments to Author:

Referee: 2

Comments to the Author(s)

Comments to the Authors:

I think the authors have made a good job to revise the ms, both in terms of the new analyses and in rewriting, and thus I have no more comments to add.

Referee: 3

Comments to the Author(s)

Dear authors,

thank you for following the recommendations. I believe that your manuscript has been substantially improved now and is a really fascinating paper. I am absolutely content with your changes and only have one minor concern left: The raw output of the Mess maps is very difficult to interpret. I would recommend showing areas of non-analogous climate hatched (e.g. in figure 4a). Areas of model uncertainty should be immediately visible to the reader, e.g. you might include them as hatched area in figure 4A.

Two other issues should be checked for grammar:

Line 616 – „Accepting the assumption...“ Thanks for including this recommend content but I believe somehow this phrase is not a complete English sentence? Please check grammar.

Line 544ff – Please check scientific species names in your references - they should be written in italics. I might have missed a few names but at least check also line 538 and 571.

Kind regards

Author's Response to Decision Letter for (RSPB-2021-1066.R1)

See Appendix B.

Decision letter (RSPB-2021-1066.R2)

07-Oct-2021

Dear Dr Warmuth

I am pleased to inform you that your manuscript entitled "Major population splits coincide with episodes of rapid climate change in a forest-dependent bird" has been accepted for publication in Proceedings B.

Data Accessibility section

Open Access

Paper charges

Sincerely,

Appendix A

Manuscript RSPB-2021-1066

Response to Reviewers

Dear Dr. Rowe,

Thank you for giving us the opportunity to submit a revised draft of the manuscript “Major population splits coincide with episodes of rapid climate change in a forest-dependent bird” for publication in *Proceedings of the Royal Society B*. We appreciate the time and effort that you and the reviewers dedicated to providing feedback on our manuscript and are grateful for the insightful comments on our paper that allowed us to improve our work. We have incorporated almost all of the suggestions made by the reviewers. Those changes are highlighted in yellow in the revised manuscript attached to this document. We are pleased to report that the results and conclusions are the same as before. Below, please find a point-by-point response to the reviewers’ comments and concerns. All line numbers refer to the revised manuscript file attached to this document.

Reviewers' Comments to the Authors:

REVIEWER 1

This paper is a welcome addition to the Ficedula Files! It cautions that accurate knowledge of mutation rates can profoundly affect divergence time estimates - that is obvious enough - but that default attribution of within-species divergences to LGM dynamics can be faulty. It is the latter caution that is timely and always worth hearing as are the analyses behind it.

A: We are glad the referee found merit in our work. We expand on the referee’s comments below.

Without wanting to clutter the text, I might suggest using the subspecies names assigned to these populations as it sets up some context along the lines of rates of differentiation between and within subspecies too. The African populations are not mentioned anywhere but they do show up in the modeling. They are recognized as *F. h. speculigera*, yes? Worth a comment somewhere, even if in the Discussion, about how they might fit in?

A: Based on molecular evidence, the African populations formerly recognised as *F. h. speculigera* are now recognised as a full species, *F. speculigera* (Atlas flycatcher) by most authorities. We discuss the nomenclature of *F. hypoleuca* in the ESM, but we agree with the reviewer that a brief mention in the ms itself would not go amiss. We now include a brief comment in the methods section (line 402ff).

Colours in K= 4 and 5 in Figure 2 are hard to make out. Do you even need to show those values?

A: we have removed the panels corresponding to K=4 and K=5 from the figure referred to by the reviewer.

Figure 3's labelling seems to have gone awry and should definitely be checked and fixed.

A: Thank you for pointing this out. We have fixed the order of the labels.

Why not refer to the subspecies taxonomy of the populations as per my comment box alongside Figure 1?

A: We have included a note on the subspecies taxonomy in the legend as suggested by the reviewer. Please note that we have had to move previous Figure 1 to the ESM in order not to exceed the page limit (now Figure S1).

And how might the African form *F. h. speculigera* fit into this story?

A: See above. We have added a sentence on the nomenclature we adhere to in the ms (methods), and discuss it further in the ESM.

Perhaps a sentence for future work in the Discussion could be added.

A: done

REVIEWER 2

This paper makes a case for how rapid climate change can lead to population splits. The divergence time was estimated by DADI using 2D-SFS for each population pairs. I suggest to complementarily infer the demographic models by incorporating all 3 major populations in one time instead of pair by pair inferences.

A: We have repeated the demographic inference using 3D-SFS as the reviewer suggested and include the output in the ESM. While the divergence times estimates of the analyses from the 3D-SFS overlap with those produced by the 2D-SFS, the confidence intervals are wider, possibly because our sample sizes are rather small for the number of parameters that need to be estimated (12 in the 3D model vs six in the 2D model). Based on this, and our confidence in the 2D models, which were extensively tested elsewhere (Warmuth & Ellegren 2019, *Mol Ecol Res*), we chose to present the parameter estimates from the 2D model in the ms, but show split times derived from the 3-population model in the ESM, Figure S6.

Line 81: I guess you used mutation rate from ref.44 rather than from ref.19 here, as same in your dadi analysis.

A: Thank you for pointing this out. We indeed meant to refer to the (former) ref.44. The inclusion of additional references has changed the overall numbering but we made sure to refer to the correct reference here.

Line 159 : Please elaborate further on this model.

A: We have included a brief description of the model (line 472ff).

Line 248-293: I guess the panels in figure 3 were erroneously arranged.

A: Thank you for pointing this out. We have fixed the order of the labels.

Line 355: change 26.5 to 26.5 – 19.

A: done

It would be much better to explain a little bit why is the previous divergence time estimation quite different from the current study. From the current descriptions, the origination of the previous divergence time is obscure.

A: We now clarify the origin of previous assumptions regarding the divergence of the Spanish lineage (line 610ff).

Figure 2A: As some overlapped circles will lead to darker colors, and it's not easy to discriminate "E" from some of the CNE populations, so it may be better to change circles to other symbols of "E" and "UK".

AU: done

REVIEWER 3

Your manuscript is well written, structured and most importantly it deals with a very interesting topic.

A: We are glad the referee found merit in our work and acknowledge the constructive comments on which we expand below.

My comments mainly focus on the ENM part of your study. Here, I see a few drawbacks that can unfortunately only be tackled by a reanalysis of the data. As the ENM part is important for your study, I believe it can be substantially improved.

A: We appreciate this comment, and hope that we have addressed the issues mentioned by the reviewer satisfactorily. Below we address each comment one by one.

I missed a more intensive engagement with ENMs (e.g. references, theoretical background, more algorithms, occurrence and background sampling should be improved etc.). I hope that my comments will help to improve your fascinating research.

A: We would like to express our thanks to the reviewer and the editor for giving us the opportunity to improve on the ENM part of our study. We have addressed all the issues highlighted by the reviewer, and are pleased to report that the results and conclusions have not changed.

Line 52 ff Introduction

Your introduction includes a short paragraph on the progress in molecular methods and you cite articles using molecular methods. However, a paragraph on ENMs is lacking and I missed at least a few standard references for SDMs/ENMs (e.g. Elith et al. 2006, *Ecography*; Guisan & Thuiller 2005, *Ecol. Let.*; Elith & Leathwick 2009, *AREES*; Engler et al 2017, *J. Avian Biol.* for avian SDMs). ENMs were also used for reconstructing the influence of Pleistocene climate fluctuations and this should not be ignored (some are perhaps better included in the discussion). You might refer to Rödger et al. 2013, *PLoSOne* (they even reconstructed ancestral niches); Peterson & Anamza 2016, *J. Avian Biol.*; Schidelko et al. 2011, *Biol J Lin. Soc.*; Schidelko et al 2013, *J. Avian Biol.* (both refer to diversity patterns), Kamp et al. 2018, *Zool. Scripta* (Phylogeography + SDM).

A: We agree with the reviewer that the introduction is missing a paragraph on ENMs. The introduction now includes a paragraph on ENMs and cites examples of their application in birds (line 393ff).

Note also that your approach assumes that the climate niche of the Pied Flycatcher is stable over time (check Engler et al. 2017 and the references therein for a brief review on niche conservatism in birds). As this is a prerequisite for your approach, it should be stated somewhere in a brief sentence.

A: We have included a sentence stating that our conclusions from the SDM assume that climate niches are stable over time (line 617).

Line 61ff

Please add a few older references. While many details were discovered during the last 25 years by new methods, the importance of the Pleistocene period for today's biota has been known or hypothesised at least for decades (e.g. see Haffer 1969 and many more).

A: We agree with the reviewer that older relevant references should not be neglected and have rectified this oversight.

Line 80

Are these data transferrable from one species to another (perhaps add a half-sentence why this is ok)?

A: We have included a sentence justifying our use of mutation rate estimates between closely related species (line 491ff.)

Line 91

Data from the east of the distribution is obviously missing. Given your focus on the Iberian Population this seems to be ok for me, but you should give a half-sentence why your sampling effort is adequate (see also comment on nomenclature in ESM).

A: We agree with the reviewer that genetic data from the east of the distribution would be ideal. We did in fact try to obtain good quality DNA from the samples we had available from this region but unfortunately both the quality and quantity of DNA from these samples were insufficient for RAD sequencing. We still believe that our sampling is adequate for what we wanted to do, namely to link major population splits with past climatic events. The absence of genetic data from the eastern half of the pied flycatcher distribution could mean that one or several more divergence events within the species are missed (though see Lehtonen et al. 2009, Mol Ecol, for a microsatellite based study showing no additional sub-structuring east of the Ural mountains). But even if this were the case, it would not invalidate the present result of the two major divergence events (for which we have solid genetic evidence) occurring during times of rapid climate change. We have included a statement to this effect in the ms (line 413ff.)

Line 188

biomod2 offers different modelling approaches, all of them have different strengths and drawbacks. Why did you focus on these four (why not taking nearly all), e.g. why did you ignore Maxent (perhaps the most prominent of the available algorithms)? I can only assume that computation time is the limiting factor but using only four approaches somehow contradicts the sense of an ensemble approach to me.

AU: We acknowledge that the different models included in biomod2 have different strengths and drawbacks, and that the ensemble approach only really makes sense when many different models are included. We chose the four models that were initially included because they were shown to outperform many complex, machine-learning based approaches when the aim is to project into new environments that are likely very different from the present-day climate (transferability). Our thinking was that a more balanced selection of model algorithms (i.e. not too many machine-learning models), would make for a better transferability. We now include seven models and find the same overall

pattern of a range collapse around 100 kya ago and a continued presence of the species in Europe south of the ice sheets during the LGM.

We appreciate that Maxent is popular and widely used, but Maxent was developed to predict the present-day distribution of species and has repeatedly been shown to have limited transferability (e.g., Phillips & Dudík, 2008; Wenger and Olden 2012, Vale et al. 2014, Fourcade et al. 2018, Fernandes et al. 2019). Furthermore, in order to produce reliable estimates, it requires occurrence data points to adequately cover the geographical distribution of a species (e.g., Li et al. 2020, Forests). We believe that this is not the case in pied flycatchers, as reliable occurrence data for this species outside of Europe are scarce. We have therefore decided to exclude Maxent from our list of models contributing to the ensemble.

Line 190/191

I must have missed it, but while I found model specifications I did not find the model evaluations (e.g. TSS, AUC) in the ESM.

A: Thank you for pointing this out. We have now included the full model evaluations in the ESM (Table S2).

Line 193

Why did you take August instead of May? Most birds should have returned by mid may, and by the end of July the species has largely left its breeding grounds as far as I know? Please correct me if I am wrong as I am most familiar with the situation in Central Europe.

In general, it is very good to see that you limit your ENM to climate from the breeding season (but you should explain to non-ornithologists that the species is migratory and not present in the area during other months in a half-sentence).

A: We appreciate this comment and have changed the breeding time to May, June and July. We have also added a sentence explaining that the species is migratory (line 514ff).

Line 200ff

From my point of view, using occurrence data from a distribution map in a single species approach is far from current standards (you mention the need for improvement in Spain but these problems could occur anywhere). I am aware that it takes some work but for a single species, actual observations should be used, e.g from GBIF and observer platforms (ornitho, observation.org; eBird shares its data with GBIF etc.).

This makes modelling more complicated (observer bias etc.) but these aspects could be tackled by appropriate background sampling etc. Using occurrences from Birdlife Polygons might be adequate for a macroecological study with a lot of species but not for a single species.

A: We re-ran the entire ENM analysis using the breeding evidence dataset collected for and curated by the EBBA2 project (50x50km²), complemented with cleaned occurrence data from gbif for the part of the distribution east of the Urals.

Background sampling (as detailed in the ESM) is also unusual and not adequate for your research questions. While a migratory species might reach a larger area than a small let's say amphibian species your background is far too large (overfitting). Better use a buffer for background sampling or sample background points from ecoregions where the species is recorded (check e.g. Guisan et al. 2014, *Trend. Ecol Evol*; Stiels et al. 2021, *JOrn* for examples).

A: We now restrict our background sampling to areas near the presence points, as recommended in Phillips et al. 2009, *Ecol Appl.* (ESM Figure S3). The area from which background points are sampled is now contained within, but smaller than, the area we project our model onto (ESM, Figure S7).

Better take 10.000 background points, and check if background points are taken in a similar way than your occurrence data (degree grid cells are larger close to the pole and smaller in temperate regions – given the latitude your data span, this should not be ignored). I think, you could use the *dismo* R package for this purpose.

A: To account for the effect of latitude on cell sizes we use the approach implemented in *dismo* as suggested by the reviewer. Due to the now smaller area available for the sampling of background points, fewer than 10,000 points were available, but the available background points (N=6585) cover all non-presence cells (ESM, Figure S4).

Please highlight areas of model uncertainty. Typically, your models extend (in different ways) beyond the training climate. It is necessary to show novel climates. You should make MESS maps (areas could be given as hatched overlays in the maps) (Elith et al. 2010, *Methods Ecol Evol*).

A: We have included mess maps for the time slices discussed in the paper as described in Elith et al. 2010, *Methods Ecol Evol* in the ESM (Figure S8).

Line 215

In the ESM you wrote that the TSS value was set to 0.8 (see below). Please check.

A: Thank you for pointing this out. It should have read 0.8. We have corrected the erroneous values.

Line 390 ff

The figure 5 on tree cover was a very interesting bonus. Reading your discussion, it would be interesting to know a bit more about deciduous vs. coniferous forests during the LGM and later on. You discuss this issue for earlier times in line 418 ff for the split of the Spanish population. However, as far as I know the British population avoids nearly any coniferous forests. In the discussion (line 390ff), you dismiss a direct link to climate fluctuations. However, indirect effects via different refugia for deciduous and coniferous forests seem to be possible? Please correct me if I am wrong. If you agree, please include a sentence on deciduous vs coniferous forest preferences for the split of the British population.

A: We included a sentence on deciduous vs coniferous forest preferences (line 661ff).

Line 405

“little is known” - But see references using SDMs mentioned above.

A: rewritten as ‘few studies have’

Line 438

Consider a brief sentence on biotic interactions with other flycatchers (e.g. Semicollared in the Caucasus which is present as refugium in your ENM). Could this have affected the observed biogeographical pattern? It is possible that the fundamental niche is underestimated your model (Soberón & Peterson 2005, Biodivers. Informatics for niche concept) – the Pied Flycatcher might simply avoid the Caucasus due to competition with its congener. As far as I understood from one of your articles (Sætre & Sæther 2010, Mol. Ecol.) competition could (at least theoretically) alter the biogeography of the Pied Flycatcher as well. Putative refugia in the Maghreb and the Caucasus are inhabited by Atlas FC and Semicollared FC. And the absence of the Pied FC today (or in the past?) in these areas might not be explained by climate factors but by competition. This is to some degree speculative but a proper model projected onto these areas can give insight into this question. Otherwise, your results must be interpreted much more carefully.

A: We include a discussion of biotic interactions with other flycatchers and how competition, rather than climate, might explain the absence of pied flycatchers in suitable niches as predicted by the models (line 719ff).

ESM

Line ESM-36

Great to see an abstract on nomenclature. Thank you! I was already wondering why *F. speculigera* is

not included in your approach but as a species on its own (basal to Collared and Pied) it makes sense to omit occurrence data within the current framework of your study.

A: Based on comments from one of the other reviewers we now mention the reason for the absence of *F. speculigera* from our occurrence data already in the ms.

Please, could you add a short sentence on *F. h. sibirica*? Your genetic sampling does not cover this subspecies but your ENM covers its distribution (which makes sense). Even if you do not want to speculate on e.g. refugia for this taxon, a brief mention seems adequate (e.g. line 91).

A: Based on molecular data from eastern Eurasian populations of *F. hypoleuca* (Lehtonen et al. 2009, Mol. Ecol.), there is no evidence to support subspecies status for *F.h.tomensis* (erroneously sometimes referred to as *F.h.sibirica* (Salvador et al. 2017, Zootaxa). We briefly mention subspecies assignments in *F. hypoleuca* and how they differ when based on molecular vs. morphological data in the method section (line 402ff), and refer to a more detailed discussion of this issue and our rationale for following a nomenclature based on molecular data in the ESM.

Line ESM-228

You wrote in the manuscript, that TSS threshold was set to 0.7. (Line 215) Which value is correct?

A: See above, this is corrected now

Appendix B

Manuscript RSPB-2021-1066

Response to Referees

Dear Dr. Rowe,

Thank you for accepting our manuscript “Major population splits coincide with episodes of rapid climate change in a forest-dependent bird” for publication in *Proceedings of the Royal Society B*. We are delighted at the positive responses our revisions have received and address the additional comments made by referee 3 below. All line numbers refer to the revised manuscript file attached to this document.

Sincerely,

Vera Warmuth and co-authors

Reviewers' Comments to the Authors:

Referee: 2

I think the authors have made a good job to revise the ms, both in terms of the new analyses and in rewriting, and thus I have no more comments to add.

A: Many thanks!

Referee: 3

Dear authors,

thank you for following the recommendations. I believe that your manuscript has been substantially improved now and is a really fascinating paper.

A: Many thanks!

I am absolutely content with your changes and only have one minor concern left: The raw output of the Mess maps is very difficult to interpret. I would recommend showing areas of non-analogous climate hatched (e.g. in figure 4a). Areas of model uncertainty should be immediately visible to the reader, e.g. you might include them as hatched area in figure 4A.

A: Thank you for this comment. We agree that it would be preferable to immediately see non-analogous climates on the main figure. However, the LGM was climatically highly dissimilar to the current climate and this is reflected in negative MESS values for the entire study area. As a compromise, we have indicated the region of the highest model uncertainty as a hatched area in Figure 4A and refer to the full set of MESS maps in the ESM.

Two other issues should be checked for grammar:

Line 616 – „Accepting the assumption...“ Thanks for including this recommend content but I believe somehow this phrase is not a complete English sentence? Please check grammar.

A: done (line 343 ff)

Line 544ff – Please check scientific species names in your references - they should be written in italics. I might have missed a few names but at least check also line 538 and 571.

A: done (lines 579, 581, 606, 688)